

# COVID-19 related messaging, beliefs, information sources, and mitigation behaviors in Virginia: a cross-sectional survey in the summer of 2020

Rachel A. Silverman[1], Danielle Short[2], Sophie Wenzel[2], Mary Ann Friesen[3] and Natalie E. Cook[2]

[1] Statistics, Virginia Polytechnic Institute and State University (Virginia Tech), Blacksburg, VA, United States of America
[2] Population Health Sciences, Virginia Polytechnic Institute and State University (Virginia Tech), Blacksburg, VA, United States of America
[3] Inova Healthsystem, Falls Church, VA, United States of America

Corresponding authors
Rachel A. Silverman,
rsilverman@vt.edu
Natalie E. Cook, necook@vt.edu

## ABSTRACT

**Background**. Conflicting messages and misleading information related to the coronavirus (COVID-19) pandemic (SARS-CoV-2) have hindered mitigation efforts. It is important that trust in evidence-based public health information be maintained to effectively continue pandemic mitigation strategies. Officials, researchers, and the public can benefit from exploring how people receive information they believe and trust, and how their beliefs influence their behaviors.

**Methods**. To gain insight and inform effective evidence-based public health messaging, we distributed an anonymous online cross-sectional survey from May to July, 2020 to Virginia residents, 18 years of age or older. Participants were surveyed about their perceptions of COVID-19, risk mitigation behaviors, messages and events they felt influenced their beliefs and behaviors, and where they obtained information that they trust. The survey also collected socio-demographic information, including gender, age, race, ethnicity, level of education, income, employment status, occupation, changes in employment due to the pandemic, political affiliation, sexual orientation, and zip code. Analyses included specific focus on the most effective behavioral measures: wearing a face mask and distancing in public.

**Results**. Among 3,488 respondents, systematic differences were observed in information sources that people trust, events that impacted beliefs and behaviors, and how behaviors changed by socio-demographics, political identity, and geography within Virginia. Characteristics significantly associated ($p < 0.025$) with not wearing a mask in public included identifying as non-Hispanic white, male, Republican political identity, younger age, lower income, not trusting national science and health organizations, believing one or more non-evidence-based messages, and residing in Southwest Virginia in logistic regression. Similar, lesser in magnitude correlations, were observed for distancing in public.

**Conclusions**. This study describes how information sources considered trustworthy vary across different populations and identities, and how these differentially correspond to beliefs and behaviors. This study can assist decision makers and the public to

improve and effectively target public health messaging related to the ongoing COVID-19 pandemic and future public health challenges in Virginia and similar jurisdictions.

## INTRODUCTION

The early days of the coronavirus (COVID-19) pandemic (SARS-CoV-2) were characterized by conflicting messaging from nearly all levels of national and international mass media and government (*Doran, 2020*; *Gaviria, Smith & PBS, 2020*; *Kolstoe, 2020*). As public health and healthcare professionals attempted to quell the growing panic with science-driven narratives, conspiracy theories, misinformation, and disinformation continued to spread through social media platforms such as Facebook, Weibo, and Twitter, often undermining or contradicting the life-saving messages that scientists were trying to communicate (*Garrett, 2020*; *Islam et al., 2020*). This issue was further compounded by long-standing health, socioeconomic, and racist inequities as well as sharp decreases in funding to state and federal health agencies in the United States (*Garrett, 2020*). Throughout the pandemic, access to and acceptance of evidence-based messaging to prevent and respond to outbreaks of coronavirus disease (COVID-19) have been inconsistent across populations and subject to politicization (*Jones et al., 2020*; *O'Shea & Ueda, 2021*). Black, Hispanic, and Indigenous populations have been historically excluded from the United States' public health and medical institutions, often suffering disproportionately from many diseases and public health challenges (*Hutchins et al., 2009*; *Krishnan, Ogunwole & Cooper, 2020*). Given that that ethnic minority, low-income, low-education, and elderly populations are also overrepresented in COVID-19 related morbidity and mortality numbers, public health officials will need to effectively reach out to and target those particular groups (*Alobuia et al., 2020*; *Holtgrave et al., 2020*; *Maroko, Nash & Pavilonis, 2020*).

Several surveys have evaluated the awareness and concern that members of the public have experienced towards COVID-19 and local, state, and national government responses (*Jones et al., 2020*; *Seale et al., 2020*). Results showed that the majority of the general population wants to hear from public health and medical officials, and are likely to trust professional sources that have self-protective and pro-social messages that focus on positive ways to protect themselves and their loved ones (*Banker & Park, 2020*; *Shelus et al., 2020*). This includes demographic groups that are considered high-risk for COVID-19, like the elderly and low-income individuals from minority groups (*Geldsetzer, 2020*; *Li et al., 2020*; *McFadden et al., 2020*).

As vaccine coverage increases at different rates globally, the public health response to COVID-19 continues to necessitate coordination at all levels of government to ensure accessible and accurate testing, contact tracing, quarantine and isolation, treatment, and mitigation measures like social or physical distancing and mask wearing (*Mobula et al., 2020*). It is important that trust in public health information be maintained for

these strategies to continue to be implemented effectively. Studies found that trust in public health officials and the information they provided allowed for successful messaging campaigns with past disease outbreaks, ranging from food safety incidents to worldwide polio vaccination campaigns (*Li et al., 2020*). As shown in past responses to foodborne disease outbreaks, demonstrating that public health measures and preventative strategies are in the best interests of the community overall is crucial to building and maintaining public trust that is essential to effective public health guidance (*Henderson et al., 2020*). High-risk populations may respond best when targeted with official messages that are consistent, credible, proactive, and also a mixture of self-focused and prosocial (*Banker & Park, 2020*).

Throughout the COVID-19 pandemic in the United States, social distancing, school closures, lockdowns, and targeted public health messaging have been sporadic and inconsistent. Many people obtain information from social media that can conflict with the messages from public health officials (*Islam et al., 2020*). In response, Facebook, Twitter, and online newspapers are now actively monitoring their own sites for inaccurate COVID-19 information that could mislead people into believing potentially dangerous rumors, stigmas, and conspiracy theories (*Islam et al., 2020*; *Krause et al., 2020*). The rapid development and rollout of COVID-19 vaccines have also been subject to false information (misinformation and disinformation) on social media platforms, with peer-networks exchanging large quantities of anti-vaccination posts that focus on adverse side-effects, misleading medical content, and unsubstantiated rumors (*Puri et al., 2020*). Similar methods have been used to undermine prior vaccine campaigns, and developing effective messaging to counter such false information will likely prove to be an important challenge for public health officials (*Dror et al., 2020*; *MacDonald & Hesitancy, 2015*).

Like many other large states, Virginia has had notable regional differences in case trends over time, with the more densely populated northern and central regions experiencing large case increases during the pandemic's initial wave in the spring of 2020, while the coastal eastern region and the more rural southwestern region experiencing their first large case increases mid-summer (*Virginia Department of Health, 2021*). Some Virginia college towns, such as Charlottesville, Blacksburg, and Harrisonburg, saw increases in local case counts when students returned in the late summer and mid-winter of 2020-21, showing that the movement of large groups of people can greatly affect community spread in less densely populated areas (*Sidersky & Sauers, 2021*; *Smith, Hwang & Binkley, 2020*). Given the continued need for effective evidence-based public health messaging, officials, researchers, and the public can benefit from exploring how people receive information they believe and trust, and how their beliefs influence their behaviors. In addition to not previously being of focus for this type of research, Virginia is a geographically and culturally diverse location, which allows for nuanced analysis of factors that influence messaging and behaviors within this state that can be generalized to other similar populations. Evaluating the differences in public health messaging and its effectiveness across difference socioeconomic and demographic groups can inform and improve future targeted messaging efforts. To gain better insight for understanding and developing effective messaging, we summarized and described the results from a cross-sectional survey administered during the

summer of 2020 to examine COVID-19 related messaging, beliefs, information sources, and mitigation behaviors among adults in Virginia. We aimed to identify correlations between messaging, behaviors, and characteristics.

Portions of this text were previously published as part of a preprint (DOI: 10.1101/2021.08.18.21262217).

## MATERIALS AND METHODS

We surveyed Virginia residents by distributing a link to complete the survey online through our professional and personal email listservs, *via* Facebook (including advertisements targeted to Virginia residents), and on flyers in select locations. Eligibility criteria included being 18 years of age or older and residing in Virginia. Participants provided electronic informed consent prior to beginning the survey questions. The survey collected socio-demographic information, including gender, age, race, ethnicity, level of education, income, employment status, occupation, changes in employment due to the pandemic, political affiliation, sexual orientation, and zip code. Participants were asked about their perceptions of COVID-19, risk mitigation behaviors, messages and events they felt influenced their beliefs and behaviors, and where they obtained information that they trust. The full survey, developed and administered using Qualtrics, is available in the supplement. To limit people from completing the survey more than once, participants were able to save and continue the survey and the *Prevent Ballot Box Stuffing* setting was selected. Responses were completely anonymous.

### Analysis strategy

We conducted exploratory analyses by calculating descriptive statistics of survey responses and investigated correlations between information sources, perceptions, beliefs, and risk mitigating behaviors related to the COVID-19 pandemic. Figures presenting these comparisons are used to visualize these comparisons and differences greater than 5% are reported. We also investigated correlates of the fundamental risk mitigating behaviors mask wearing and social/physical distancing in unadjusted and adjusted analyses using logistic regression with robust variance estimates. Statistical significance was taken at the 0.025 level to account for these two primary outcomes using a conservative Bonferroni adjustment and a nominal type I error rate of 0.05. We adjusted for race, political identity, gender, age group, income, reporting national science and health organizations as an information source, believing in alternative messages, and living in southwest Virginia to identify the independent effects of these characteristics on risk mitigation behaviors. These variables were selected a priori as known correlates of COVID-19 beliefs and incidence (*Alobuia et al., 2020*; *Azlan et al., 2020*; *Barari et al., 2020*; *Benham et al., 2021*; *Bonyan et al., 2020*; *Christensen et al., 2020*; *McCaffery et al., 2020*; *McFadden et al., 2020*; *Sherman et al., 2021*; *Wolf et al., 2020*). Survey responses were excluded only if none of the questions beyond eligibility were answered.

All analyses were conducted using Stata/SE 16.1 and Microsoft Excel. This work was conducted by the Community and Collaborative subgroup of the integrated Translational Health Research Institute of Virginia (iTHRIV), a collaboration between Virginia Tech,

University of Virginia, Inova, and Carilion Clinic. This study was approved by the Virginia Tech institutional Review Board (IRB number: 20-353) and the Inova Institutional Review Board (IRB number: U20 05-4056), prior to initiation of study activities at the respective sites.

# RESULTS

## Respondent characteristics

The survey was open from May 19th to July 19th, 2020. Of the 3,694 individuals who started the survey, 3,678 (99.6%) self-reported as eligible and of these 190 (5%) did not answer any survey questions and were excluded. Of the remaining 3,488 respondents 3,367 (97%) fully completed the survey. Of the 3,488 included in this analysis, 70% completed the survey in May, 21% in June, and 9% in July of 2020. Participants were represented throughout Virginia (Fig. 1), with the largest numbers of respondents residing in Montgomery County (home of Virginia Tech), Loudoun and Fairfax Counties (near Washington DC, home of Inova), and Wise County (home of UVA Wise), reflecting sites where survey recruitment began.

Sociodemographic characteristics are presented in Table 1. Briefly, 78% were women, 3% were Hispanic, 83% were non-Hispanic white, 5% were Black, 3% were Asian, and 87% identified as heterosexual or straight. Six percent were 18–24 years old, 6% were 25-29, 16% were 30-39, 19% were 40-49, 20% were 50-59, 20% were 60-69, and 9% were 70 or older. Most (94%) had completed at least some college or other post-high school education or training. Forty-three percent reported annual household income at least $100,000, 11% between $80,000 and $100,000, 13% between $60,000 and $80,000, 12% between $40,000 and $60,000, and 11% less than $40,000. Forty-six percent of respondents identified as Democrat, 13% as Republican, 22% as independent, and 13% said other or no-preference. Employment status was not mutually exclusive and 58% had full-time employment, 12% had part-time employment, 16% were retired, 5% were students, and 4% were unemployed. Eighteen percent of respondents reported a loss of or reduced employment or income, while 70% reported no change in their employment status as a result of COVID-19.

## Trusted information sources

All but 16 respondents (0.5%) answered the question: "Where do you get information that you trust about coronavirus/COVID-19?" Of these ($n = 3,472$), most reported national science and health organizations (85%) as a trusted source for COVID-19 information and over 50% of respondents reported state/local health departments (75%), healthcare professionals (74%), and online news sources (55%) as a trusted source. Information sources reported by less than half of respondents included family and friends (26%), faith leader (4%), local TV news (34%), national TV news (49%), printed newspapers (20%), radio (20%), social media (22%), local government leaders (46%), and federal government leaders (22%). Only 2% of these respondents reported not following any COVID-19 updates (Fig. 2A).

More women than men received information they trusted from local government leaders (48% *vs.* 41%) and more men than women received information they trusted from

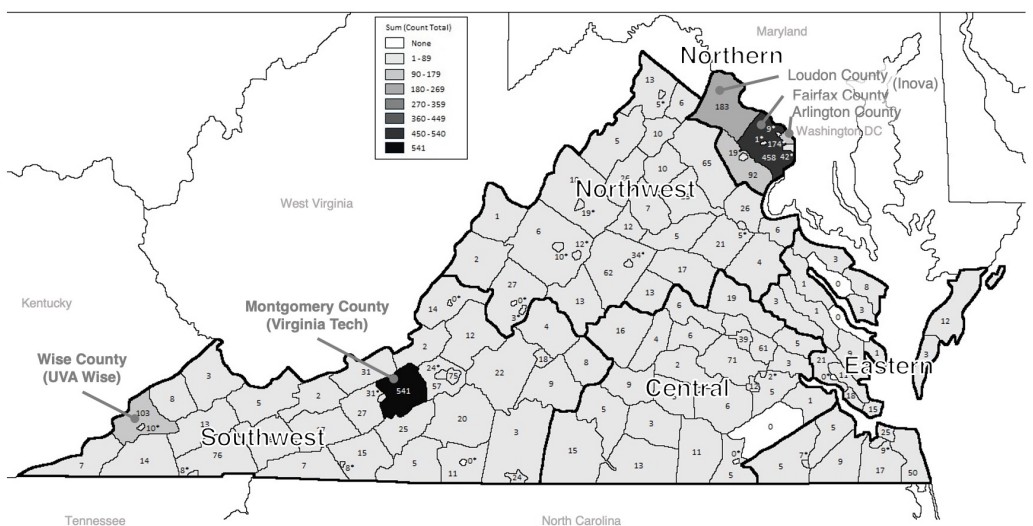

**Figure 1 Map of survey respondents by county in Virginia (*N* = 3,307).** Number of respondents residing in each county in Virginia. This figure was generated using ArcGIS.

family/friends (31% *vs.* 25%), and federal government leaders (26% *vs.* 20%) (Fig. 2B). Young adults age 18-24 were more likely than those of all older ages combined to receive trusted information from family and friends (39% *vs.* 25%) and social media (31% *vs.* 21%) and less likely from printed news (6% *vs.* 21%), radio (10% *vs.* 21%), and local government leaders (36% *vs.* 47%) (Fig. 2C). Slightly less non-Hispanic white and Asian respondents received trusted information from faith leaders (4% and 3%, respectively) than other races including 9% of Black and 10% of multiracial respondents. Non-white respondents were also more likely than white respondents to receive information from local TV news (42% *vs.* 33%), and social media (27% *vs.* 21%) (Fig. 2D). White respondents were more likely than non-white respondents to receive information from printed local newspapers (21% *vs.* 14%), national science and health organizations (87% *vs.* 82%), or state and local health departments (77% *vs.* 71%) (Fig. 2D). More Democrats than Republicans received information they trusted from national science and health organizations (93% *vs.* 72%, respectively), State or local health department (83% *vs.* 62%), online news (65% *vs.* 41%), national TV news (55% *vs.* 43%), local government leaders (54% *vs.* 46%), printed newspapers (45% *vs.* 13%) and radio (26% *vs.* 14%) (Fig. 3A). More Republicans than Democrats received information they trusted from federal government leaders (46% *vs.* 13%), faith leaders (6% *vs.* 3%), or did not follow coronavirus/COVID-19 updates (5% *vs.* 0.3%). Similar proportions by political identity received information from local TV news, family/friends, healthcare-professionals, and social media.

More of those with than without a college degree received information from local printed newspapers (22% *vs.* 13%), radio (22% *vs.* 15%), online news (58% *vs.* 44%), local government leaders (47% *vs.* 43%), national science and health organizations (89% *vs.* 74%), and State or local health departments (79% *vs.* 64%) (Fig. 3B). More of those without than with a college degree received information from local faith leaders (6%

**Table 1  Characteristics of survey respondents ($N = 3,488$).**

| Category | Subcategory | n (%) | Category | Subcategory | n (%) |
|---|---|---|---|---|---|
| Age group years | 18–24 | 207 (6%) | Household income | Less than $20,000 | 129 (4%) |
| | 25–29 | 215 (6%) | | $20,000 to $39,999 | 268 (8%) |
| | 30–39 | 562 (16%) | | $40,000 to $59,999 | 401 (12%) |
| | 40–49 | 646 (19%) | | $60,000 to $79,999 | 465 (13%) |
| | 50–59 | 710 (20%) | | $80,000 to $99,999 | 387 (11%) |
| | 60–69 | 688 (20%) | | $100,000 or more | 1,500 (43%) |
| | 70+ | 315 (9%) | | Missing | 338 (10%) |
| Region | Central | 346 (10%) | Political Affiliation | Republican | 467 (13%) |
| | Eastern | 234 (7%) | | Democrat | 1,608 (46%) |
| | Northern | 977 (28%) | | Independent | 761 (22%) |
| | Northwest | 474 (14%) | | Other | 150 (4%) |
| | Southwest | 1,276 (36%) | | No preference | 320 (9%) |
| | Missing | 181 (5%) | | Missing | 182 (5%) |
| Race / Ethnicity[a] | American Indian or Alaska Native | 7 (<1%) | | Less than high school degree | 5 (<1%) |
| | Asian | 93 (3%) | | High school or GED | 93 (3%) |
| | Black | 159 (5%) | | Trade school or Associate degree | 241 (7%) |
| | Middle Eastern | 7 (<1%) | | Some college (no degree) | 333 (10%) |
| | White, non-Hispanic | 2,879 (83%) | Education | Bachelor's degree | 1,095 (31%) |
| | Multiracial | 68 (2%) | | Master's degree | 1,059 (30%) |
| | Hispanic | 104 (3%) | | Doctoral or professional degree | 534 (15%) |
| | Missing Race/Ethnicity | 180 (5%) | | Missing | 128 (4%) |
| Gender | Women | 2,724 (78%) | | Heterosexual or straight | 3,051 (87%) |
| | Men | 613 (18%) | Sexual Orientation | LGBTQ+ | 209 (6%) |
| | Other | 20 (<1%) | | Missing | 228 (7%) |
| | Missing | 131 (4%) | | Full Time | 2,017 (58%) |
| | No change | 2,446 (70%) | | Part Time | 421 (12%) |
| | Permanently lost primary income source | 47 (1%) | | Seeking Opportunities | 110 (3%) |
| | Temporarily lost primary income source | 138 (4%) | | Retired | 568 (16%) |
| How has employment changed as a result of the pandemic? | Gained employment/income | 53 (2%) | Current Employment[a] | Not working due to disability | 33 (1%) |
| | Employment/Income was reduced | 441 (13%) | | Student | 185 (5%) |
| | Other | 180 (5%) | | Not Employed | 150 (4%) |
| | Missing | 183 (5%) | | Other | 114 (3%) |
| | | | | Missing | 140 (4%) |

**Notes.**

[a]Race/ethnicity and employment categories are not mutually exclusive.

*vs.* 4%), local TV news (38% *vs.* 33%), and federal government leaders (26% *vs.* 20%). More higher- than middle- and lower-income individuals received information from local printed newspapers (23%, 18%, 15%), online news (58%, 55%, 51%), local government leaders (49%, 43%, 45%), national science and health organizations (89%, 84%, 83%), and State or local health departments (80%, 77%, 70%) (Fig. 3C). Differences in information sources were also observed across regions indicating some heterogeneity geographically (Fig. 3D). Other differences presented in Figs. 2 and 3 were less than 5%.

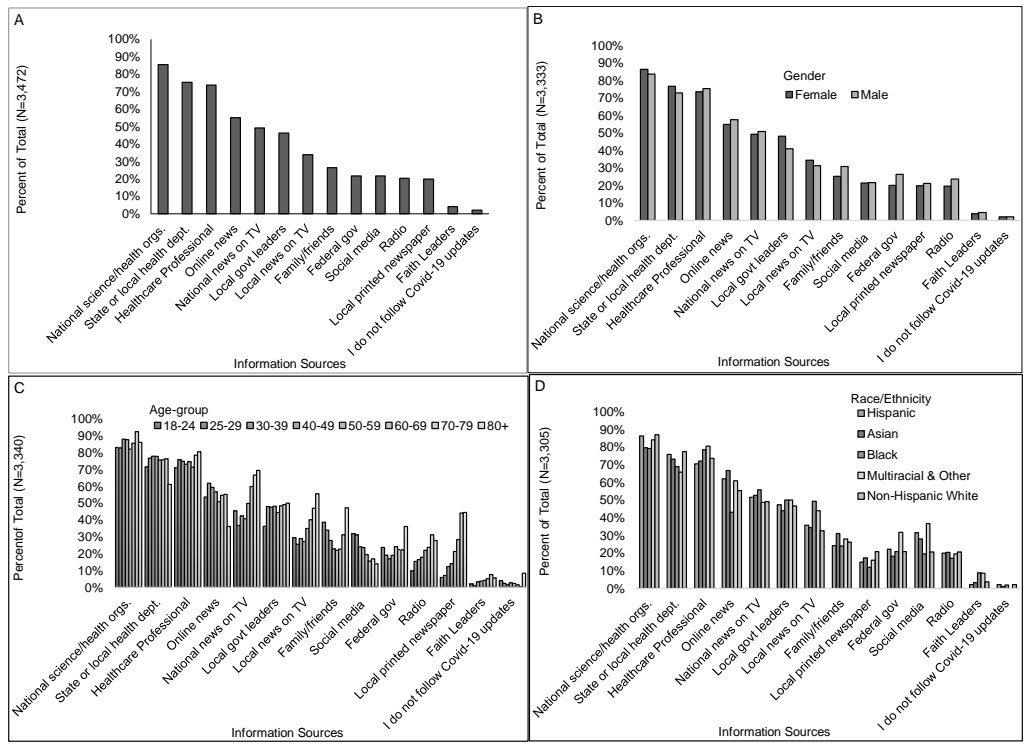

**Figure 2** **Where respondents received information that they trust about COVID-19, overall and by select participant characteristics (gender, age-group, and race/ethnicity).** Survey responses to the question: "Where do you get information that you trust about coronavirus/COVID-19? (Check all that apply)" for (A) all respondents, and by (B) gender, (C) age-group, and (D) race/ethnicity.

## Perceptions & beliefs related to COVID-19

Twenty-three percent of respondents reported being "very worried" about catching COVID-19 and 34% were "very worried" about experiencing severe disease or complications if they were to catch COVID-19. Most respondents considered COVID-19 to be very serious (82%) or somewhat serious (13%). Demographic differences greater than 5% among those considered COVID-19 to be very serious were observed between women *vs.* men (85% *vs.* 77%), Democrats *vs.* Republicans *vs.* others (95% *vs.* 61% *vs.* 78%), those 60 or more years old *vs.* those under 60 years old (91% *vs.* 81%), LGTBQ+ *vs.* heterosexual (89% *vs.* 83%), and higher-income *vs.* lower-income (86% reporting $100,000 *vs.* 78% making less than $20,000), and those with *vs.* without a college degree (86% *vs.* 75%). Other demographic characteristics were not <5% different in terms of perceived seriousness of COVID-19. Differences by race and ethnicity were also observed with 91% of Asian, 87% of Black and Hispanic, 84% of multiracial and white, non-Hispanic respondents reporting they believed COVID-19 to be very serious.

Among those who answered the question: "Which if any of the following impacted your belief that COVID-19 was serious" (*n* = 3,371), more than half selected hearing about COVID-19 in other countries (77%) or other states (73%), public-school closings (69%), the governor recommending a stay-at-home order (51%), mandating a stay-at-home order
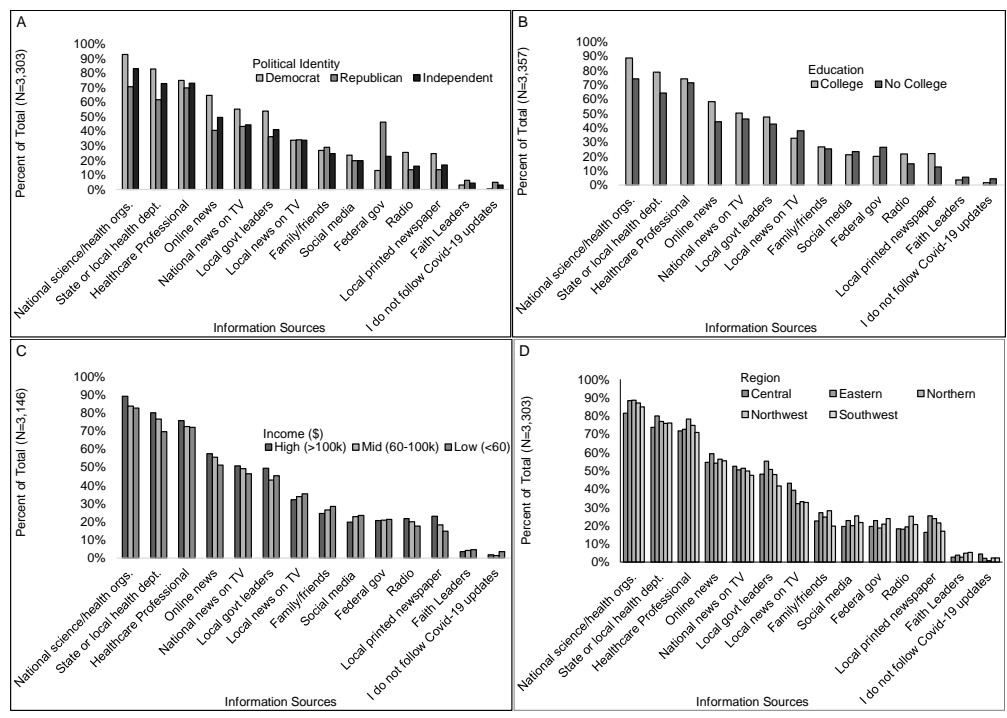

**Figure 3** **Where respondents received information that they trust about COVID-19, by select participant characteristics (education level, income, and region).** Survey responses to the question: "Where do you get information that you trust about coronavirus/COVID-19? (Check all that apply)" by (A) political identity, (B) education level, (C) income level, and (D) Virginia region.

(65%), and declaring a state of emergency (68%), the CDC recommending that everyone wear a face mask in public (67%), restaurant dining rooms shutting down, (58%), and sporting events being canceled or postponed (52%) (Fig. 4A). Less than half selected becoming sick (3%) or knowing someone who became sick (30%), being high-risk or living with some-one who is high-risk for severe disease (40%), starting to work from home (35%), being laid off or losing their job (5%), when stores began limiting purchases of essential items (42%), when religious services were moved online (40%), and when a public figure tested positive (17%). Two percent of those who responded to this question said they did not think COVID-19 is serious, and 11% selected some other reason not listed in the survey.

Among those who answered the question: "Which if any of the following impacted your belief that COVID-19 was serious," women were more likely than men to state someone they knew became sick (31% *vs.* 26%), being or living with someone high-risk (42% *vs.* 32%), public schools closing (71% *vs.* 63%), the governor declaring a state of emergency (70% *vs.* 63%), the CDC recommending face masks (69% *vs.* 63%), hearing about it in other countries (78% *vs.* 75%) and states (74% *vs.* 69%), stores limiting purchases (43% *vs.* 38%), religious services gong online (41% *vs.* 35%), and the governor recommending a stay at home order (52% *vs.* 47%) and mandating stay at home order (68% *vs* 57%) (Fig. 4B). Young adults were more likely than all other age-groups combined to select restaurant

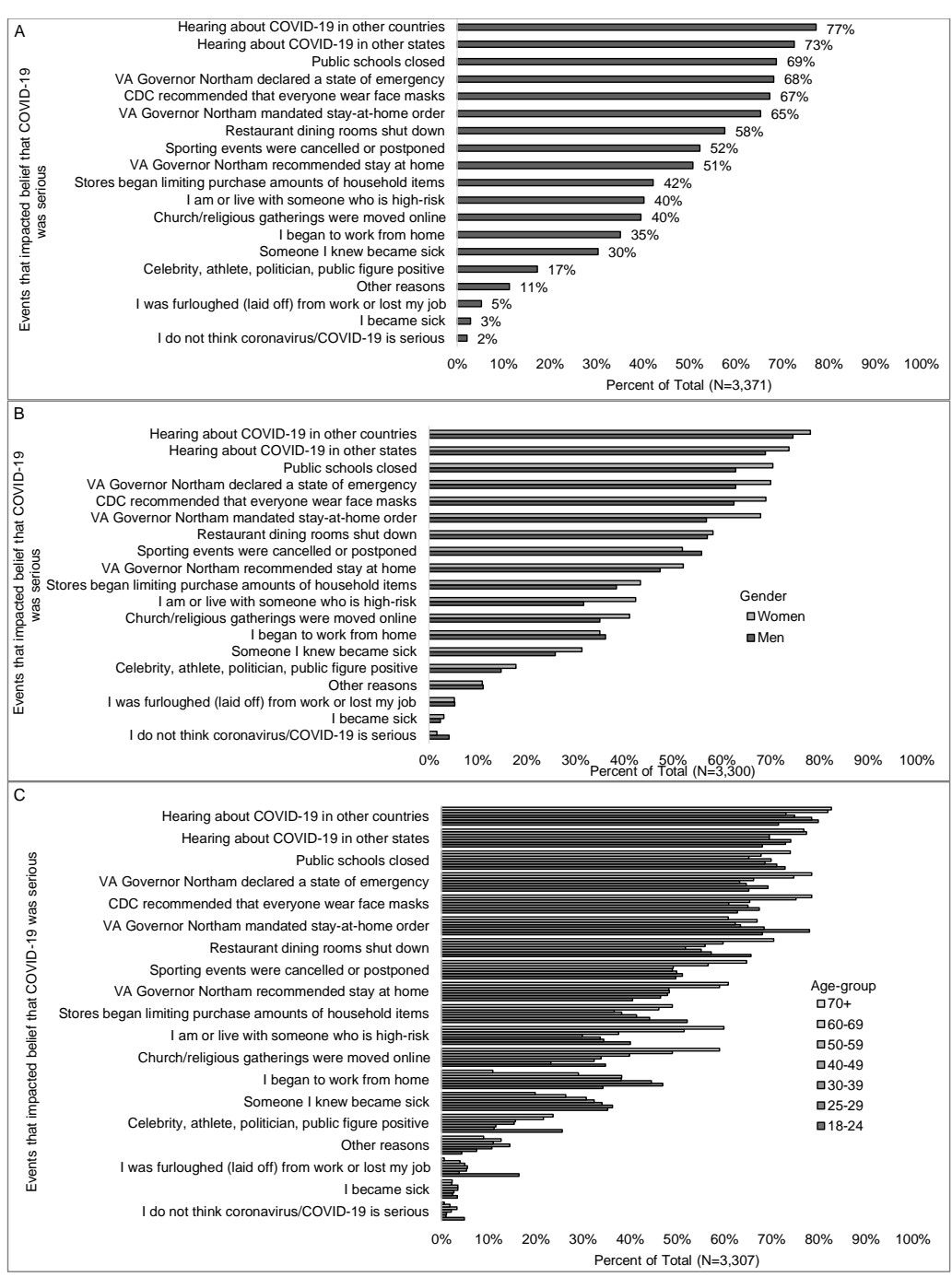

**Figure 4** Events impacting belief that COVID-19 was serious, overall and by select participant characteristics (total, gender, and age-group). Survey responses to the question: "Which (if any) of the following have affected whether or not you think the coronavirus/COVID-19 is serious? (Check all that apply)" for (A) all respondents, and by (B) gender, and (C) age-group.

dining rooms (66% *vs.* 57%), being furloughed (laid off) or losing their job (16% *vs.* 4%), stores limiting purchases, (52% *vs.* 42%), and a public figure testing positive (26% *vs.* 17%) and less likely to select hearing about COVID-19 in other countries (72% *vs.* 78%), the governor *recommending* a stay-at-home order (41% *vs.* 52%), though this is not true for when the governor *mandated* the stay-at-home order (68% *vs* 66%) compared to other age groups (Fig. 4C). Non-white respondents were more likely than white respondents to select personally becoming sick (6% *vs.* 2%) or someone they knew becoming sick (45% *vs.* 28%), stores limiting purchases (52% *vs.* 41%), religious services moving online (47% *vs.* 39%), and a public figure testing positive (23% *vs.* 17%) (Fig. 5A). Democrats were more likely than Republicans to select knowing someone who was sick (32% *vs.* 24%), public schools closing (74% *vs.* 62%), restaurants closing (61% *vs.* 53%), the governor declaring a state of emergency (80% *vs.* 45%), starting to work from home (40% *vs.* 29%), the CDC recommending face masks (76% *vs.* 53%), hearing about COVID-19 in other countries (86% *vs.* 59%) and states (81% *vs.* 58%), stores limiting purchases (46% *vs.* 38%), sporting events being canceled or postponed (56% *vs.* 48%) and a public figure testing positive (20% *vs.* 13%) (Fig. 5B). More Republicans than Democrats selected religious services moving online (45% *vs.* 37%). More of those with than without a college degree reported their belief was impacted by the governor declaring a state of emergency (70% *vs.* 63%), beginning to work from home (38% *vs.* 24%), the CDC recommending face masks (69% *vs.* 63%), hearing about COVID-19 in other countries (81% *vs.* 65%) or states (75% *vs.* 65%), and sporting events being cancelled (53% *vs.* 49%) (Fig. 5C). More of those without than with a college degree reported their belief was impacted by becoming sick (5% *vs.* 3%), being or living with someone who is high-risk (47% *vs.* 39%), being furloughed (laid off) from work (9% *vs.* 4%), stores limiting purchases (46% *vs.* 41%) and religious services moving online (46% *vs.* 38%).

Fewer higher-income (>$100,000) than middle-income ($60,000–$99,999) and lower-income (<$60,0000) reported their belief was impacted by being or living with someone who is high-risk (37%, 41%, 44%), being furloughed (laid off) from work or losing their job (4%, 5%, 9%), stores limiting purchases of essential items (37%, 46%, 48%), religious gatherings being moved online (36%, 42%, 41%), a celebrity getting testing positive for COVID-19 (16%, 16%, 22%), and/or the governor recommending a stay at home order (49%, 55%, 52%). More higher- than middle- and lower-income reported beginning to work from home (39%, 35%, 32%), and hearing about COVID-19 in other countries (80%, 77%, 75%) (Fig. 6A). Differences in what impacted the beliefs that COVID-19 was serious were also observed across regions (Fig. 6B) for becoming sick, knowing someone who became sick, being or living with someone at high risk, beginning to work from home, and sporting events being cancelled or postponed.

## Evidence based and alternative messages related to COVID-19

Over 80% of respondents reported that one or more of the following evidence-based messages impacted their beliefs and/or behaviors: "the coronavirus is highly contagious;" "stay home stay safe;" "stay home, save lives, slow the spread;" "practice social distancing;" "don't touch your face;" and "wash your hands for at least 20 seconds" (Fig. 7). Of note for

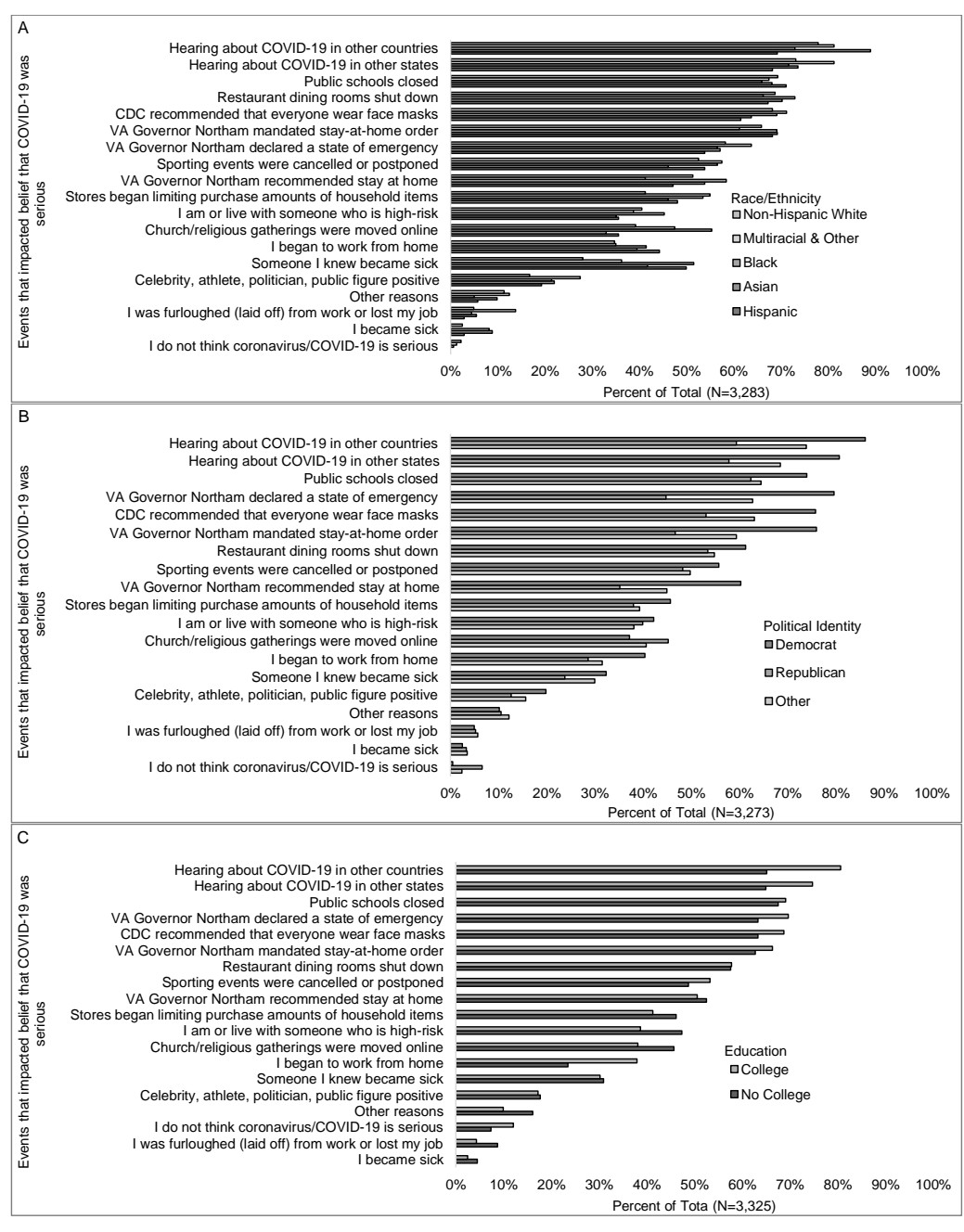

**Figure 5** **Events impacting belief that COVID-19 was serious, by select participant characteristics (race/ethnicity, political identity, and education level).** Survey responses to the question: "Which (if any) of the following have affected whether or not you think the coronavirus/COVID-19 is serious? (Check all that apply)" by (A) race/ethnicity, (B) political identity, and (C) education level.

focused analyses on alternative messages, 12% percent of respondents reported believing in one or more of the following: COVID-19 "was developed as a bioweapon" (6%), "was developed to lower social security payments to seniors" (1%), "is a sign of the

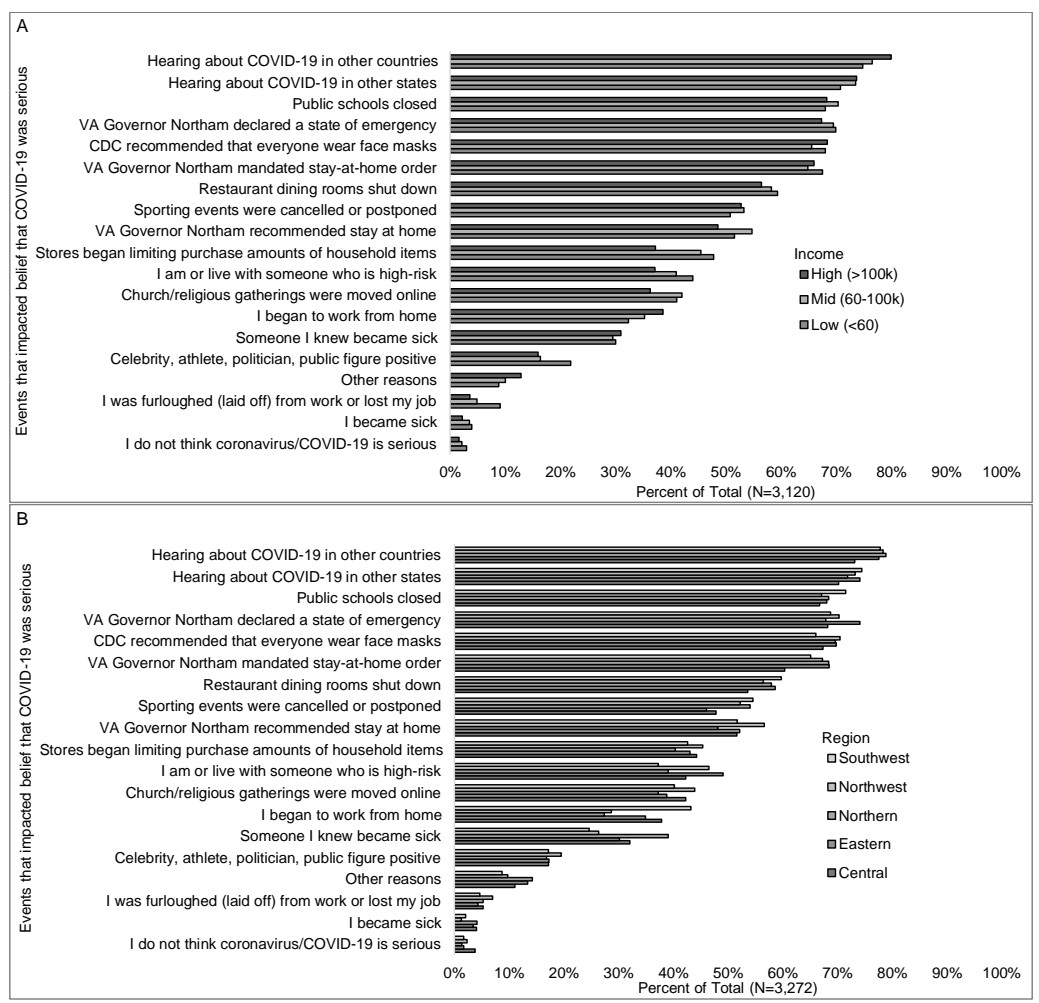

**Figure 6** Events impacting belief that COVID-19 was serious, by select participant characteristics (income level and region). Survey responses to the question: ''Which (if any) of the following have affected whether or not you think the coronavirus/COVID-19 is serious? (Check all that apply)'' by (A) income level, and (B) Virginia region (6b).

apocalypse/end times'' (3%), ''is a hoax'' (1%), ''can be treated with natural remedies'' (3%), ''was developed for population control'' (3%), and/or ''was developed to increase sales of cleaning supplies'' (4%).

## Correlations between alternative messages, demographics, and information sources

The same proportion (12%) of men and women (Fig. 8A), more young adults *vs.* older ages combined (22% *vs.* 12%), with the lowest among those 70 years old and greater (5%) (Fig. 8B), less of those who identified as non-Hispanic white (10%) compared to other races/ethnicities including 37% of Black, 22% of multiracial, 21% of Hispanic, and 15% of Asian (Fig. 8C), more Republicans (24%) than Democrats (7%) and others (15%) (Fig. 8D) believed in one of more alternative messages. The percent of those who believed in one of

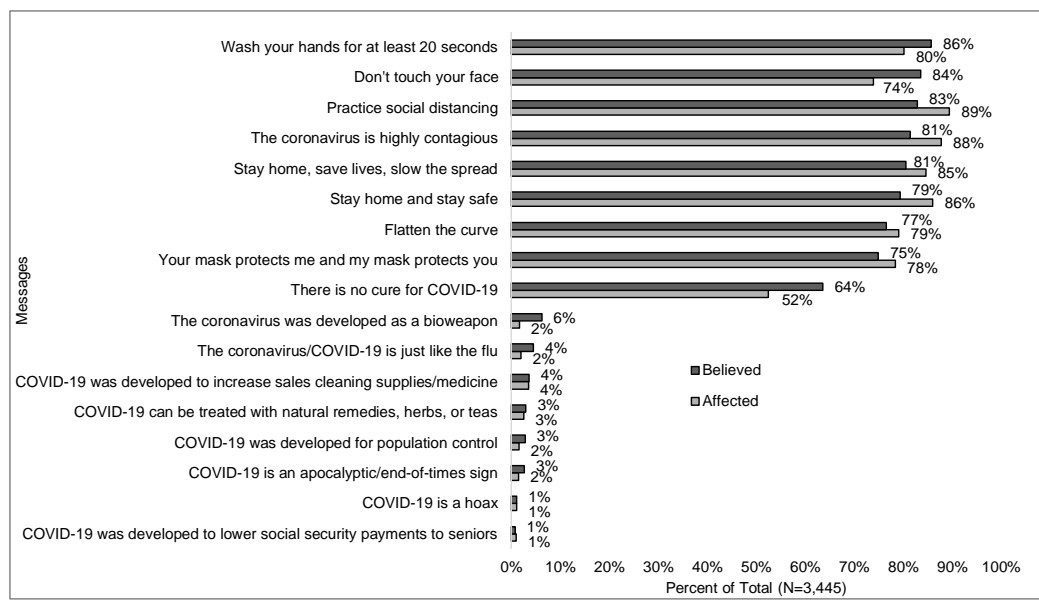

**Figure 7** Messages that respondents believe and/or affected their behaviors ($N = 3,445$). Survey responses to the question: "The following messages are related to the coronavirus/COVID-19 (not all are true). Please check all that apply if you have heard, believe, and/or changed your behavior based on each message" for all respondents.

more alternative messages decreased with increasing education (29% of high-school degree or less to 6% of those with a doctoral degree) (Fig. 8E), and with increasing income level (9%, 13%, 17%) (Fig. 8F). More of those in Central Virginia than other regions (20% *vs.* <14%) (Fig. 8G) believed in one or more alternative messages.

Those who believed in an alternative message were more likely than those who did not to receive trusted information from family and friends (32% *vs.* 26%), a faith leader (8% *vs.* 3%), local TV news (38% *vs.* 33%), social media (29% *vs* 21%), federal government leaders (33% *vs.* 20%), or report not following COVID-19 updates (Fig. 8H). Those who did not believe any alternative messages were more likely than those who did to receive trusted information from a healthcare professional (74% *vs.* 69%), local newspapers (21% *vs.* 13%), radio (21% *vs.* 15%), online news (57% *vs,* 45%), local government leaders (48% *vs.* 38%), national science and health organizations (88% *vs.* 67%), and state or local health departments (78% *vs.* 59%).

## Risk mitigation behavior changes

Ninety-eight percent of respondents completed the questions about changes in behaviors and 98% of those reported changing their behavior in some way in response to the pandemic (Fig. 9). More than half of respondents reported one or more of the following behavior changes: practicing social/physical distancing (95%), wearing a mask when in public (90%), washing hands more often (90%), shopping for groceries and other essentials less often (86%), washing hands for 20 s (86%), being more careful not to touch their face in public (82%) and/or with unwashed hands (80%), using hand sanitizer more often (79%),

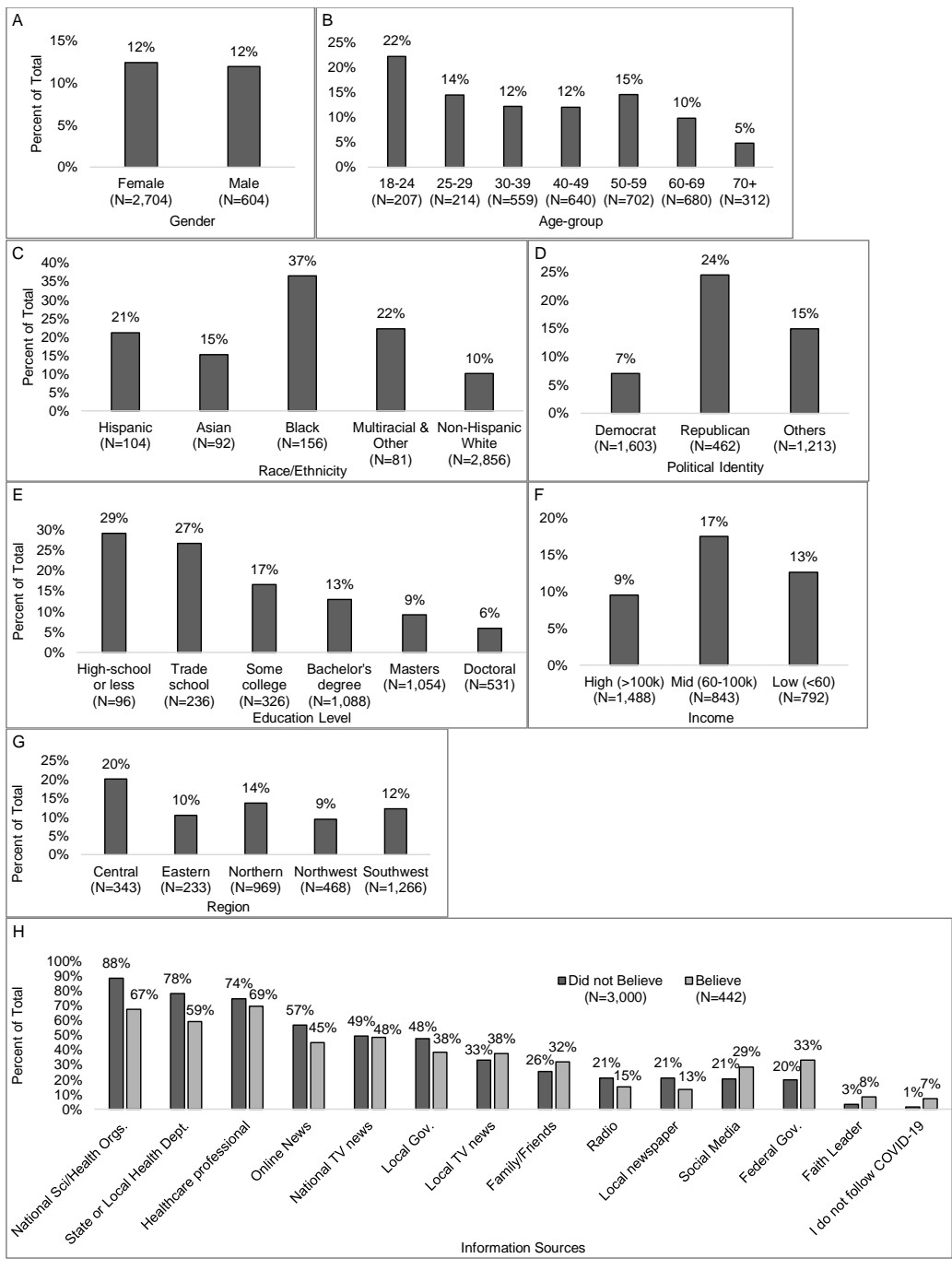

**Figure 8  Percent that believed one or more alternative messages\*, by participant characteristics.** Percent of respondents who selected they believed in one or more alternative message when answering the question: "The following messages are related to the coronavirus/COVID-19 (not all are true). Please check all that apply if you have heard, believe, and/or changed your behavior based on each message" by (A) gender, (B) age-group, (C) race/ethnicity, (D) political identity, (E) education level, (F) income level, and (G) Virginia region. \*Alternative messages response options include: COVID-19 "was developed as a bioweapon," "was developed to lower social security payments to seniors," "is a sign of the apocalypse/end times," "is a hoax," "can be treated with natural remedies," "was developed for population control," and "was developed to increase sales of cleaning supplies.".

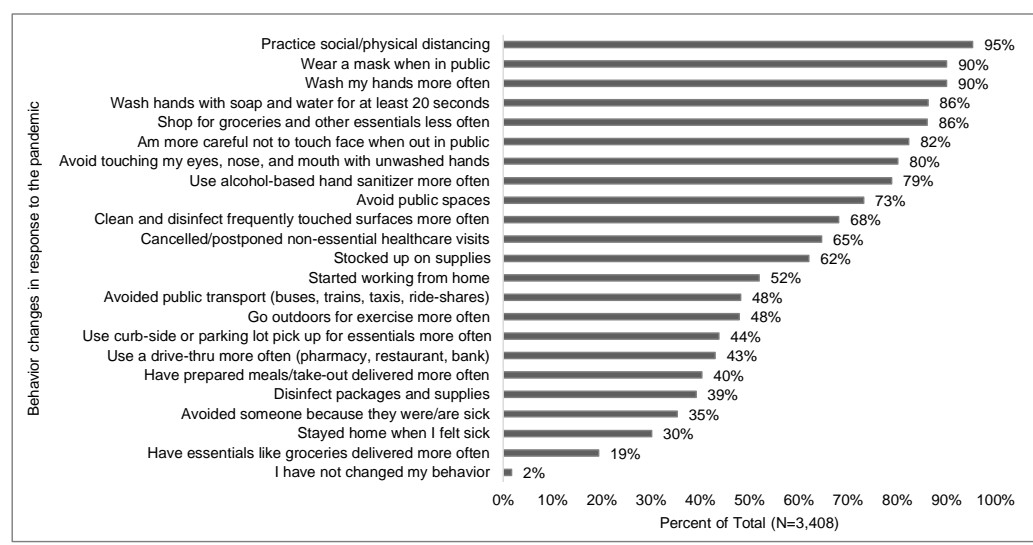

**Figure 9** **Behavior changes reported in response to the pandemic.** Survey responses to the question: "How (if at all) have you changed your behavior in response to the coronavirus/COVID-19? (Check all that apply)" for all respondents.

avoiding public spaces (73%), cleaning frequently touched surfaces (68%), stocking up on supplies (62%), and started working from home (52%).

Wearing a mask in public was reported by more women than men (92% *vs.* 84%) (Fig. 10A), and the proportion of respondents who reported wearing a mask increased by age from 80% of those 18–24 years old to 95% of those 70 years and older (Fig. 10B). Non-Hispanic White respondents were less likely to report mask wearing (90%) compared to Hispanic (95%), Asian (95%), and Black (94%) respondents (Fig. 10C). Additionally, more Democrats than Republicans and others (97%, 77%, and 87%, respectively) (Fig. 10D) reported wearing a mask, and mask use increased with greater education from 76% of those with a high-school education or less to 94% of those with a doctoral degree (Fig. 10E), and more higher- than middle- and lower-income (92%, 89%, 87%) (Fig. 10F). Those in Southwest Virginia reported less mask wearers (86%) than other regions (91%–95%) (Fig. 10G). Distancing was more common than masking in all groups but showed similar demographic trends as wearing a mask.

More of those who reported wearing *vs.* not wearing a mask reporting national health and science organizations (89% *vs.* 58%), state or local health departments (78% *vs.* 51%), health care professional (75% *vs* 64%), online news (57% *vs.* 38%), local government leaders (49% *vs.* 24%), local TV news (35% *vs.* 26%), local newspapers (21% *vs.* 6%), and radio (21% *vs.* 13%) as a trusted source of information (Fig. 11A). A smaller proportion of those who reported wearing *vs.* not wearing a mask reported the federal government as a trusted source (21% *vs.* 28%) or not following COVID-19 information (1% *vs.* 12%). Similar trends were observed for distancing (Fig. 11B).

In adjusted logistic analyses, we found that the odds of reporting not wearing a mask in public was greater than their comparative groups for those living in Southwest Virginia *vs.*

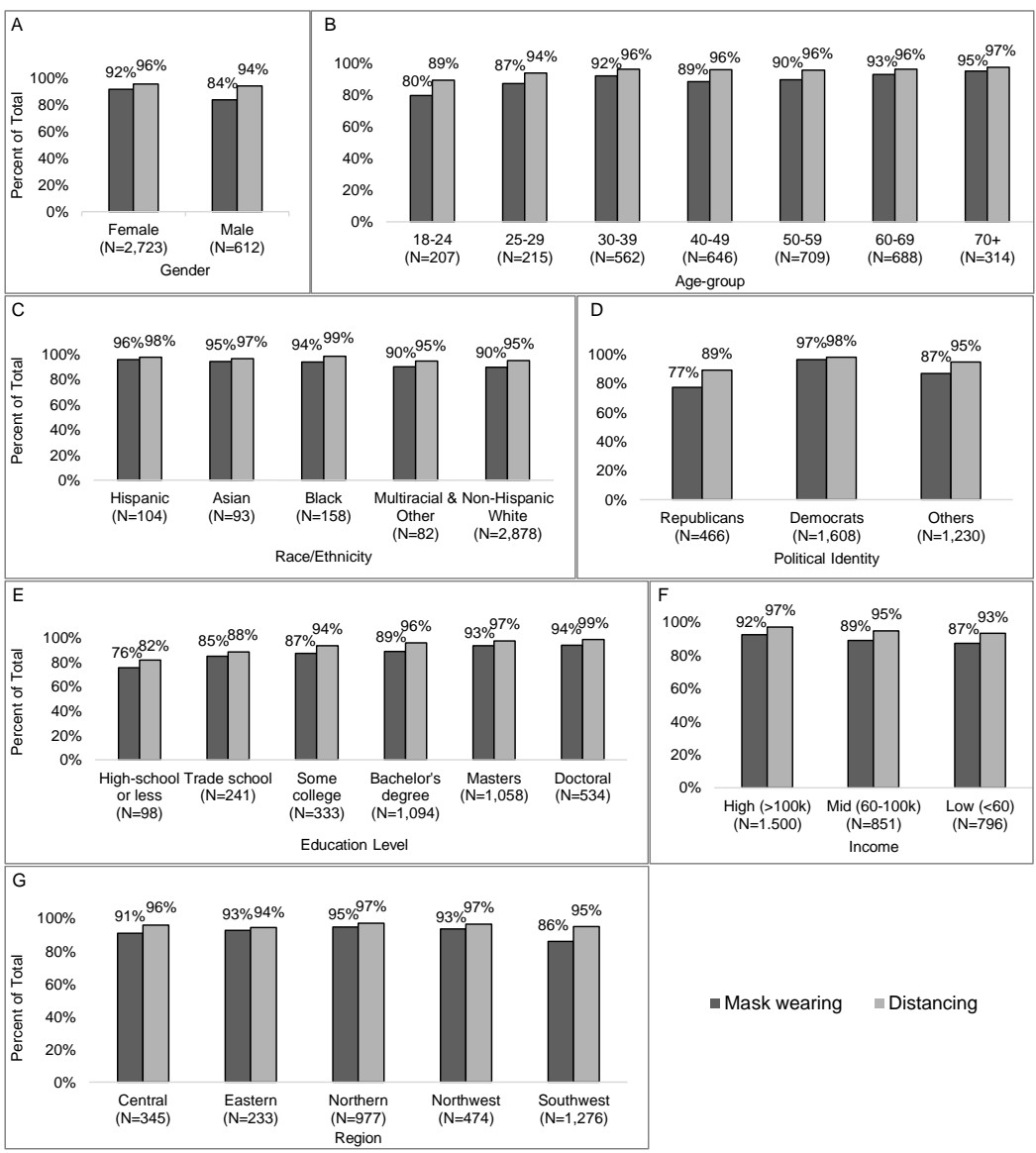

**Figure 10 Participants reporting masking and distancing in public by characteristics (gender, age-group, race/ethnicity, political identity), education level, income, Virginia region).** Survey responses to the question: "How (if at all) have you changed your behavior in response to the coronavirus/COVID- 19? (Check all that apply)." Percent of respondents reporting mask wearing and distancing by (A) gender, (B) age-group, (C) race/ethnicity, (D) political identity, (E) education level, (F) income level, and (G) Virginia region.

all other regions combined (odds ratio [OR] = 1.95; 95% CI = 1.48, 2.58; $p$-value < 0.001), men *vs.* women (OR = 1.85; 95% CI = 1.36, 2.51; $p$-value < 0.001), young adults *vs.* other age-groups combined (OR = 2.42; 95% CI = 1.54, 3.81, $p$-value < 0.001), and those with household income under $100,000 *vs.* those with income at least $100,000 (OR = 1.41; 95% CI = 1.06, 1.89; $p$-value = 0.020) (Table 2). The odds of reporting not wearing a mask in public was greater for those identifying as a Republican *vs.* Democrat (OR =5.42; 95% CI

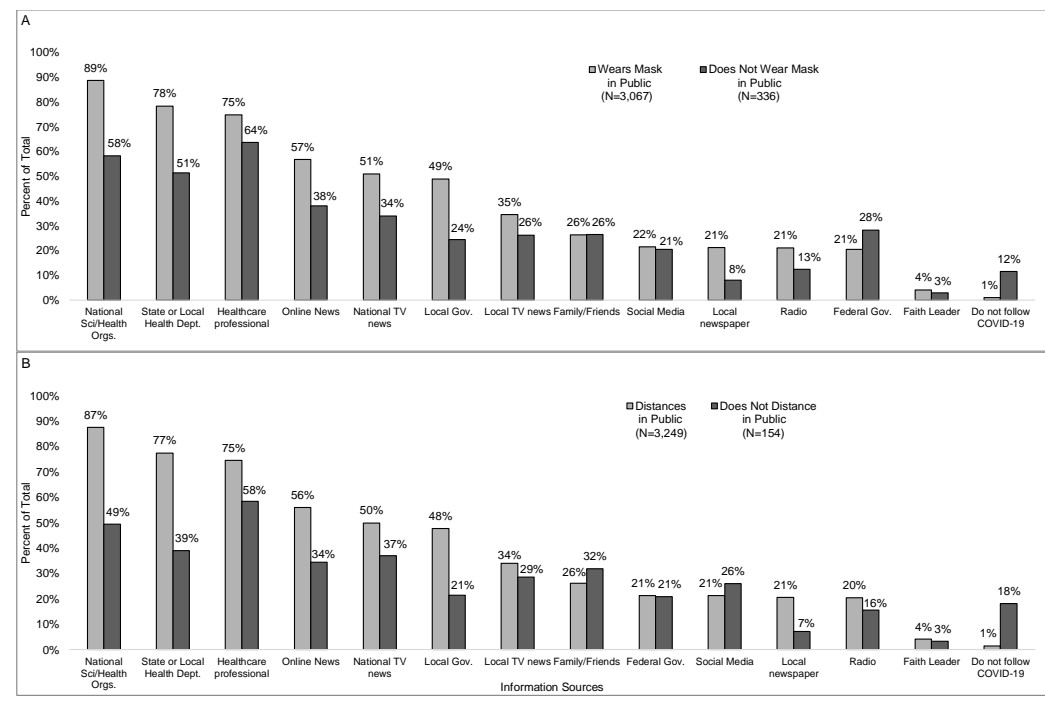

**Figure 11 Percent of respondents reporting an information source as trustworthy by if they reported wearing a mask and distancing or not in public.** Survey responses to the question: "How (if at all) have you changed your behavior in response to the coronavirus/COVID-19? (Check all that apply)." Percent of respondents reporting an information source as trustworthy by if they reported (A) wearing or not wearing a mask in public and by (B) distancing or not distancing in public.

= 3.63, 8.09; *p*-value < 0.001), those who did not *vs.* did report national science and health organization(s) as a trusted information source (OR = 3.16; 95% CI = 2.21, 4.51; *p*-value < 0.001), those who believed one or more alternative messages *vs.* not believing in any (OR = 2.09; 95% CI = 1.48, 2.94; *p*-value < 0.001). Non-Hispanic Whites were less likely to report wearing a mask *vs.* other races combined in unadjusted analysis (OR = 1.63; 95% CI = 1.09, 2.46; *p*-value = 0.018), but not in unadjusted analyses (OR = 1.68; 95% CI = 1.04, 2.71; *p*-value =0.035). Not having a college degree was associated with not wearing a mask in unadjusted analyses (OR = 1.98; 95% CI = 1.54, 2.55; *p*-value < 0.001), but not in adjusted analyses (OR = 0.98; 95% CI = 0.71, 1.35; *p*-value = 0.897). All other associations were statistically significant (*p* < 0.025) in both adjusted and unadjusted analyses. Odds ratios for distancing showed similar associations as for masking, but at a smaller magnitude. Region, gender, race/ethnicity, education, and income were not statistically significant in adjusted analyses for distancing.

## DISCUSSION

In our sample of adults residing in Virginia, we found many differences in where people received information that they trust, what they believed, and how their behaviors changed in response to the COVID-19 pandemic by socio-demographics, political identity, and

Silverman et al. (2024), *PeerJ*, DOI 10.7717/peerj.16714

Peer*J*

**Table 2  Odds ratios (ORs) of reporting not wearing masks in public and not practicing social/physical distancing using logistic regression with robust standard errors ($N = 3,307$).**

| Variables | Not wearing masks in public | | Not practicing social/physical distancing | |
|---|---|---|---|---|
| | Unadjusted OR (95% CI); *p*-value | Adjusted[*] OR (95% CI); *p*-value | Unadjusted OR (95% CI); *p*-value | Adjusted[*] OR (95% CI); *p*-value |
| Southwest *vs.* all other regions | 2.33 (1.84, 2.96); **<0.001** | 1.95 (1.48, 2.58); **<0.001** | 1.44 (1.02, 2.03); 0.037 | 1.13 (0.75, 1.70); 0.567 |
| Men *vs.* women | 2.14 (1.66, 2.76); **<0.001** | 1.85 (1.36, 2.51); **<0.001** | 1.48 (1.01, 2.18); 0.046 | 1.32 (0.82, 2.15); 0.255 |
| 18-24 *vs.* all other age-groups | 2.59 (1.90, 3.71); **<0.001** | 2.42 (1.54, 3.81); **<0.001** | 2.89 (1.79, 4.66); **<0.001** | 2.34 (1.34, 4.11); **0.003** |
| Non-Hispanic White *vs.* other races | 1.63 (1.09, 2.46); **0.018** | 1.68 (1.04, 2.71); 0.035 | 1.66 (0.91, 3.01); 0.099 | 1.83 (0.92, 3.63); 0.083 |
| No college degree *vs.* college degree | 1.98 (1.54, 2.55); **<0.001** | 0.98 (0.71, 1.35); 0.897 | 3.52 (2.52, 4.92); **<0.001** | 1.46 (0.96, 2.25); 0.080 |
| Less than $100,000 *vs.* $100,000 or more | 1.67 (1.31, 2.13); **<0.001** | 1.41 (1.06, 1.89); **0.020** | 2.11 (1.47, 3.03); **<0.001** | 1.62 (1.03, 2.54); 0.035 |
| Political Identity | | | | |
| Democrat | Ref | Ref | Ref | Ref |
| Republican | 8.79 (6.18, 12.48); **<0.001** | 5.42 (3.63, 8.09); **<0.001** | 6.92 (4.31, 11.1); **<0.001** | 3.53 (2.06, 6.07); **<0.001** |
| Independent, other, no preference | 4.44 (3.21, 6.13); **<0.001** | 3.16 (2.21, 4.51); **<0.001** | 3.15 (2.01, 4.93); **<0.001** | 1.79 (1.07, 3.01); 0.026 |
| Not reporting national science and health organizations as trusted source of information | 5.21 (4.14, 6.54); **<0.001** | 3.45 (2.57, 4.66); **<0.001** | 5.57 (4.20, 7.41); **<0.001** | 2.96 (1.95, 4.49); **<0.001** |
| Belief in alternative message(s) | 3.39 (2,65, 4.33); **<0.001** | 2.09 (1.48, 2.94); **<0.001** | 4.17 (3.09, 5.64); **<0.001** | 2.65 (1.66, 4.23); **<0.001** |

Notes.

*Adjusted models include all variables listed in the table ($N = 2,992$ with complete information for all variables). *P*-values < 0.025 are in bold.

geography within Virginia. Respondents who identified as non-Hispanic white, men, Republican, other political identity, younger age, income <$100,000, did not report national science and health organizations as a trusted source, reported believing an alternative message, and/or living in Southwest Virginia had greater odds of not wearing a mask than their comparative groups in both unadjusted and adjusted logistic regression. Differences were also observed for physical distancing for these same variables, but at a lower magnitude as distancing was more likely than masking across all groups so differences were less pronounced.

The most consistently listed trusted information sources included national health and science organizations like the NIH and CDC, state and local health departments, and healthcare professionals. Additionally, CDC and gubernatorial recommendations and, more strongly, mandates, were often reported as strong influencers of COVID-19 related beliefs and correlated with masking and distancing in public. These results emphasize the importance of these entities to communicate accurate and timely information and responsibly issue recommendations and mandates to mitigate the pandemic.

Our study was subject to several limitations. First, complete demographic and socioeconomic information was missing from 9% of respondents included in this study. Second, the political identity response options were limited to Republican, Democrat, independent, and other, resulting in individuals identifying as "independent" and "other" being grouped together, although these individuals may hold extremely diverse political views. Finally, this internet-based convenience sample is not representative of the generalized Virginia population: females, non-Hispanic white, those with at least a bachelor's degree, those with a higher income (*United States Census Bureau, 2021*), and democrats were overrepresented (*Pew Research Center, 2014*; *Virginia Department of Elections, 2023*). However, while the summary descriptive statistics are not representative of the target population of Virginia residents overall, we were able to make valid comparisons between subgroups thanks to our large sample size, many of which have been observed similarly in less detailed surveys conducted in the United States and internationally, discussed in more detail below (*Azlan et al., 2020*; *Barari et al., 2020*; *Baum et al., 2020*; *Benham et al., 2021*; *Bonyan et al., 2020*; *Carey, 2021*; *Christensen et al., 2020*; *Lee et al., 2020*; *MacDonald & Hesitancy, 2015*; *McCaffery et al., 2020*; *O'Shea & Ueda, 2021*; *Roozenbeek et al., 2020*; *Sherman et al., 2021*; *Wolf et al., 2020*). Additionally, because this is a cross-sectional survey, our results may not reflect conditions at other time points given that the survey was conducted in the summer of 2020. Data collection began on May 19th, 2020 during the Governor's executive order that directed Virginians to stay home except for essential services, bans crowds of more than 10 people, closed recreation, entertainment, and personal care businesses; and limited restaurants to only takeout and delivery services only (*Office of the Governor, 2020a*; *Office of the Governor, 2020b*). This was just prior to the racial justice protests that began on May 26th in Minneapolis and continued throughout the United States (*Taylor, 2021*). People's behavior may have been altered based on their cost-benefit analyses of COVID-19 risk and the risks associated with racial injustice over the course of our data collection period (*Godoy, 2020*; *Huang & Aubrey, 2020*). Finally, our survey did not investigate an important feature of messaging observed

in multiple other studies: consistent messaging focusing on positive ways to cope with lockdowns and other COVID-19 mitigation measures were more effective than messaging focused only on compliance in promoting long-term behavioral changes like staying at home, mask-wearing, social distancing, and hand-washing (*Azlan et al., 2020*; *Barari et al., 2020*; *Benham et al., 2021*; *Wolf et al., 2020*).

Other cross-sectional survey studies from early in the COVID-19 pandemic (spring to summer 2020) found similar overall results describing the link between evidence-based messaging and behaviors (*Alobuia et al., 2020*; *Azlan et al., 2020*; *Barari et al., 2020*; *Benham et al., 2021*; *Bonyan et al., 2020*; *Christensen et al., 2020*; *McCaffery et al., 2020*; *Sherman et al., 2021*; *Wolf et al., 2020*). For example, studies in Australia, Malaysia, Italy, Canada, the United Kingdom, Arab countries, and other regions of the United States found that evidence-based COVID-19 messaging significantly influenced respondents' beliefs and risk mitigation behaviors. Studies also showed that while older adults were generally more concerned and had higher anxiety levels about potential COVID-19 infection, they were also less concerned than younger adults about the short- and long-term economic instabilities caused by the pandemic (*Barari et al., 2020*; *Benham et al., 2021*; *Bonyan et al., 2020*; *Carey, 2021*; *Christensen et al., 2020*; *Wolf et al., 2020*). Other studies also found that women, racial/ethnic minorities, and those with lower socioeconomic status experienced more COVID-19 anxieties compared to men, ethnic majorities, and those of higher socioeconomic status (*Alobuia et al., 2020*; *Christensen et al., 2020*; *McCaffery et al., 2020*; *Wolf et al., 2020*). In multiple countries, people identifying as politically conservative and those with lower health literacy and education level were more likely to report not following recommended COVID-19 precautions and believing that people were overreacting (*Azlan et al., 2020*; *Bonyan et al., 2020*; *Christensen et al., 2020*; *McCaffery et al., 2020*; *Roozenbeek et al., 2020*). People identifying as politically liberal and those with higher health literacy and education were more likely to follow public health guidelines and believe that their governments were not doing enough to stop the pandemic (*Bonyan et al., 2020*; *Christensen et al., 2020*; *McCaffery et al., 2020*; *Wolf et al., 2020*). Multiple studies showed that false information exposure and beliefs were consistently higher among younger people, ethnic minorities, and those who identified as politically conservative (*Baum et al., 2020*; *Christensen et al., 2020*; *Lee et al., 2020*; *McCaffery et al., 2020*; *Roozenbeek et al., 2020*). Several preliminary studies have also shown that false information and mistrust in government entities and/or the vaccine development process are major contributing factors to vaccine hesitancy, especially among minority populations and people with low education levels, socioeconomic status, and low perceived risk of contracting COVID-19 (*Gatwood et al., 2021*; *Khubchandani et al., 2021*; *Troiano & Nardi, 2021*). Our study supports and adds to this knowledge describing how information sources considered trustworthy vary across these different population believes and behaviors. By focusing on Virginia, we were able to collect a large amount of data quickly during a key moment early in the pandemic across a geographically and culturally diverse area within the United States.

This study can assist decision makers and the public in developing more effective, targeted, public health messaging for both the ongoing COVID-19 pandemic and for future public health challenges in Virginia and similar settings in the United States. Future

studies could include quantitative subgroup, subregion, and qualitative analyses to enhance our understanding of the nuances related to designing effective public health messaging.

## ACKNOWLEDGEMENTS

We thank the other members of our study team: Kristin Miller, Emily Wells, Kathy Hosig, Amy Smith, Amanda Nguyen, Molly Roberts, Teace Markwalter, and Kristina Jiles for their assistance with survey distribution and recruitment. Thank you to Luela Ba and Becky Willis for helping make tables and figures. We thank the survey respondents for their participation.

### Funding

Research reported in this publication/presentation/work was supported by the National Center for Advancing Translational Sciences of the National Institutes of Health under Award Number UL1TR003015. The content is solely the responsibility of the authors and does not necessarily represent the official views of the National Institutes of Health. The funders had no role in study design, data collection and analysis, decision to publish, or preparation of the manuscript.

### Grant Disclosures

The following grant information was disclosed by the authors:
The National Center for Advancing Translational Sciences of the National Institutes of Health: UL1TR003015.

### Competing Interests

All authors declare there are no competing interests.

### Author Contributions

- Rachel A. Silverman conceived and designed the study, analyzed the data, prepared figures and/or tables, authored or reviewed drafts of the article, and approved the final draft.
- Danielle Short conceived and designed the study, analyzed the data, prepared figures and/or tables, authored or reviewed drafts of the article, and approved the final draft.
- Sophie Wenzel conceived and designed the study, authored or reviewed drafts of the article, and approved the final draft.
- Mary Ann Friesen conceived and designed the study, authored or reviewed drafts of the article, and approved the final draft.
- Natalie E. Cook conceived and designed the study, authored or reviewed drafts of the article, and approved the final draft.

### Human Ethics

The following information was supplied relating to ethical approvals (i.e., approving body and any reference numbers):

This study was approved by the Virginia Tech institutional Review Board (IRB number: 20-353) and the Inova Institutional Review Board (IRB number: U20 05-4056).

## Data Availability

The survey data and code used in our analyses are available in the Supplemental File.

Per IRB approval, we are unable to share geographic information smaller than state level, so all geographic variables have been omitted from the public dataset provided.

## Supplemental Information

Supplemental information for this article can be found online at http://dx.doi.org/10.7717/peerj.16714#supplemental-information.

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
