# Peer review of "COVID-19 related messaging, beliefs, information sources, and mitigation behaviors in Virginia: a cross-sectional survey in the summer of 2020"

_PeerJ, doi:10.7717/peerj.16714_

## Round 0.1 · original submission · Major Revisions

Your paper has been revised by 3 independent reviewers. As indicated in their in-depth analyses, your work needs a profound revision.

Reviewer 1 ·

Basic reporting

No comment

Experimental design

Introduction
In the introduction section, the authors do not clearly define their research question or illustrate a gap in the literature that this paper seeks to fill. Although the authors discuss some the COVID-19 trends specific to Virginia in lines 99 through 107, this section would be strengthened by explicitly stating, if it is in fact the case, that no existing research has looked at the patterns the authors are investigating in Virginia and stating a clear corresponding research question. This section would also be improved if, in addition to clarifying the gap in the literature and research question, the authors included a rationale for their study. Why is it important to understand these differences in Virginia specifically, when (as noted in the discussion) many of the observed associations have been documented by other studies looking at different populations?

Methods
The description of the methods includes sufficient information, and a cross-sectional survey is an appropriate study design to examine the associations of interest.

Validity of the findings

The data are provided and appear to be robust, statistically sound, and controlled. However, conclusions were not clearly linked to a research question. Specific comments on the results and discussion section are below:

Results
The results section read like a laundry list of every statistically significant result the authors found. Because the results section included so many results, it was unclear what, if any, story the authors are trying to tell. It was not clear why the gender breakdowns for every subsection of the survey were included, as the authors did not discuss gender as a meaningful predictor of any of their variables of interest in the introduction. Although the authors indicate in the introduction that ethnic minority status, income, education, and age are related to COVID-19 morbidity and mortality (lines 60 to 63), the authors could again reference this at the end of the introduction or in the methods section so that the reader is reminded why these breakdowns are important. Reporting so many results often resulted in sentences that were not clear; for example, the sentence spanning lines 324 to 329 contains so many results that are not written in parallel structure that it is difficult to follow. Similarly, the sentence spanning lines 295 to 300 also contains too many results and is difficult to read. Further, there was no attention drawn to which reported differences were particularly large in magnitude or noteworthy. Drawing attention to noteworthy differences (e.g. there were greater differences in trusted sources between Republicans and Democrats than between men and women) would make the results more compelling and interesting. Specific results of note could be highlighted in the results and further discussed in the discussion section.

Although the authors acknowledge in the discussion section that they used a convenience sample, which is not representative of the general Virginia population (lines 375 to 377), it would be useful if they also included a sentence on how their sample differed from the Virginia population (e.g. the sample is more highly educated than the general Virginia population) to give the summary descriptive statistics some context. This sentence could be added to the second paragraph of the respondent characteristics section.

Discussion
Similar to my comments on the introduction and results section, the discussion section could be improved by better conveying why the study findings are interesting and important. The fact that information sources, perceptions and beliefs about COVID, belief in misinformation, and engaging in risk mitigation behavior vary by sociodemographic characteristics and political identity has been well-documented. As currently written, the discussion section seems to indicate that the current study confirms existing knowledge. If, as the authors claim, the study adds to existing knowledge, they should more explicitly highlight which of their results are new, noteworthy, or unique to the context of Virginia. Additionally, the discussion would be strengthened if the authors were more specific on how the results of the study can assist in developing more effective health messaging (lines 426 to 428). For example, do the results indicate which channels are most effective for delivering health messages to different populations? If so, that should be clearly stated.

Also in the discussion, the authors indicate that their study describes how information sources considered trustworthy…impact beliefs and behaviors (lines 424 to 425). Because this is a cross-sectional study, “impact” is too strong of causal language to describe the association between trusted information sources and message beliefs and behavior.

Additional comments

The authors should be more consistent with their language regarding misinformation. In the introduction section, the authors refer to both misinformation (e.g. line 93) and disinformation (e.g. lines 90 and 97), but do not define the difference between the two. I suggest the authors define both terms, as is standard in the misinformation literature, and ensure that they use them consistently with their definitions. If the authors’ use of the two terms stems from the language used by the references the authors cite, providing definitions of the two terms would still be helpful. Additionally, in the results section, the authors refer to alternative messages, and in the abstract they refer to them as non-evidence-based messages. I suggest that the authors define what they mean by alternative messages in the text and use consistent language in the text and abstract.

Reviewer 2 ·

Basic reporting

no comment

Experimental design

1. I think the author(s) may add some details about the representativeness of the selected sample (3000+). Albeit it seems generalizable in terms of sample size, but it will add value if they delineate that this sample size is within the threshold of sample size formula e.g., Morgan Table or add G-Power analysis.

2. Author (s) may add a sub-heading of Analysis strategy at Line 127.

Validity of the findings

1. The paper is well-written and interesting results are reported in the study. However, the managerial implications of studies can be expanded (see Line 426). These "This study can assist decision makers and the public in developing more effective public health messaging for both the ongoing COVID-19 pandemic"...Author(s) may add more discussion on this interesting aspect of the results. For instance, what kind of messages suit the diverse public (age group/ political identity, etc.).
2. The targeted messages should be generated by public health organizations and media (what I perceive) etc.

Additional comments

Overall, the paper provides a novel description of the understudied population by selecting a unique context. The study is timely and can be beneficial for future planning to handle crisis situations.
Good Luck.

Reviewer 3 ·

Basic reporting

• In the framing in the intro, politicization also impacted the effectiveness of messaging w/r/t the covid-19 pandemic: the political right framed things differently than the political left, and this had implications for how different parts of the public understood and reacted to the pandemic (and how different parts of the public were more versus less likely to trust relevant entities and/or adopt misinformation). This should be included alongside the factors listed (misinfo, disparities, and funding issues), as this also fits in with the trust framing you include.
• I am confused as to what the first limitation is saying – what do you mean by “conducted numerous comparisons and did not adjust for multiple comparisons?”
• There are several grammar issues throughout. Here are some but not all issues:
a. In the Results part of the abstract/overview, it should be “male” and not “men” (since you are describing characteristics and not nouns).
b. On page 7 line 189 should have Non-white as plural, same with newspaper on line 192 on the same page.
c. On the first discussion page, line 366: “Differences in physical difference were…” – I think this was supposed to be “Differences in physical distance…”

Experimental design

• I like the general framing of this paper: that the results can be employed to inform public messaging strategies.
• This journal (PeerJ) is particular about having a well-defined research question which is not explicitly stated in the first couple pages of the manuscript (alongside a clear gap in knowledge that the research is filling – which seems to be understanding how people from Virginia vary in information, demographic, and source trust when evaluating their likelihood of engaging in mask wearing/social distancing so messaging can be better informed/targeted). The closest I can find to a question is on page 2 lines 108-110, but it isn't a question.
• In the Discussion you discuss limitations of survey collection and potential biased estimates due to oversampling of women, undersampling of racial minorities, etc., which is good. But there are some pretty serious biases to the survey data. One additional step would be to construct and use survey weights based on the known demographics of Virginia and then doing the analyses with the weighted data.
• In the methods section you need to be clearer about what your dependent variables are and how they are measured/coded for the regressions, and about some variable coding in general for replication.
• In terms of ethics, for the ‘alternative’ messages, what was the specific question wording, and did respondents receive a debrief on this? Exposure to this misinformation via a survey may potentially cause respondents to believe this misinformation.
• In the Discussion, include more commentary on how the timing of the survey relative to other parts of the pandemic may have potentially impacted results. For instance, were cases in VA high or low during this time? What were the state policies at this time? Etc.

Validity of the findings

• The Discussion section does not circle back to the main question (since one was not posed in the first place).
• In the Discussion, you say that your results are similar to other surveys from early in the COVID-19 pandemic, but then the next sentence compares your results to effective messaging impacting beliefs and risk mitigation behaviors, which is not what your study is doing. Precisely clarify the link you are making here – I think the intention is to say that the demographic findings can be used for effective messaging? - or use more relevant comparisons (such as comparisons with Kaiser Family Foundation) and THEN more clearly talk about how these correlational studies relate to messaging studies.
o This section reads more like a literature review and feels tangential. Instead, this section should more clearly relate your findings to potential strategies for messaging that can be used.

Additional comments

• Overall I think this has the potential to be informative in the way the authors say it is, but substantial revisions are needed.
• In the overview/abstract, be more specific about which mitigation factors are being considered (i.e., mask wearing and distancing).
• The respondent characteristics should be listed under methods rather than in the results section (at least, this is the norm in my field).

---

## Round 0.2 · accepted · Accept

Thank you for revising the original manuscript in accordance with the reviewers criticisms.